# Neuroanatomy in a middle Cambrian mollisoniid and the ancestral nervous system organization of chelicerates

Javier Ortega-Hernández [1✉], Rudy Lerosey-Aubril[1], Sarah R. Losso[1] & James C. Weaver[2]

Recent years have witnessed a steady increase in reports of fossilized nervous tissues among Cambrian total-group euarthropods, which allow reconstructing the early evolutionary history of these animals. Here, we describe the central nervous system of the stem-group chelicerate *Mollisonia symmetrica* from the mid-Cambrian Burgess Shale. The fossilized neurological anatomy of *M. symmetrica* includes optic nerves connected to a pair of lateral eyes, a putative condensed cephalic synganglion, and a metameric ventral nerve cord. Each trunk tergite is associated with a condensed ganglion bearing lateral segmental nerves, and linked by longitudinal connectives. The nervous system is preserved as reflective carbonaceous films underneath the phosphatized digestive tract. Our results suggest that *M. symmetrica* illustrates the ancestral organization of stem-group Chelicerata before the evolution of the derived neuroanatomical characters observed in Cambrian megacheirans and extant representatives. Our findings reveal a conflict between the phylogenetic signals provided by neuroanatomical and appendicular data, which we interpret as evidence of mosaic evolution in the chelicerate stem-lineage.

[1] Museum of Comparative Zoology and Department of Organismic and Evolutionary Biology, Harvard University, Cambridge, MA 02138, USA. [2] Wyss Institute for Biologically Inspired Engineering, Harvard University, 60 Oxford Street, Cambridge, MA 02138, USA. ✉email: jortegahernandez@fas.harvard.edu

Exceptionally preserved fossils represent a valuable source of morphological information that allows us to reconstruct the origin and early evolutionary history of disparate animal groups that inhabit the modern biosphere[1–4]. The study of extinct soft-bodied organisms from marine deposits has become fundamental for understanding the sudden appearance of animal macrofossils during the Cambrian Explosion, and the increasing availability of material from different deposits reveals a broader spectrum of exceptional preservation than traditionally considered possible. Burgess Shale-type (BST) sites—defined by the preservation of non-biomineralizing organisms as carbonaceous compressions in marine shale[2,5,6] demonstrate that in addition to external features such as eyes and limbs[7], various aspects of the internal anatomy may also become fossilized and reveal substantial morphological details. For example, guts commonly occur as either carbonaceous compressions[5,8] or three-dimensional mineralized replicates[9–12], as are seldom muscle fibers[12–14]. An increasing number of studies published within the last decade have also reported the preservation of nervous systems in Cambrian macrofossils[7,15–24], challenging the notion that brains and nerves are too delicate to be captured in the fossil record[4,25].

With a few exceptions[21,26,27], most instances of exceptionally preserved nervous systems in Cambrian BST deposits are restricted to members of total-group Euarthropoda, including gilled lobopodians[20], radiodonts[17], fuxianhuiids[15,19,28], megacheirans[16,22,24], and bivalved forms[18,29,30]. These records encompass a broad phylogenetic sampling, which allows reconstructing the complex evolutionary history of the central nervous system (CNS) in this phylum, clarifying the segmental origin of euarthropod head structures, and illuminating the ancestral neuroanatomical organization of extant representatives[28,31]. Yet, further discoveries of palaeoneuroanatomical data in other euarthropod groups are still critically needed for comprehensively understanding the early macroevolution of the CNS in this megadiverse phylum, and testing competing phylogenetic hypotheses.

Here, we describe the CNS organization in the mollisoniid euarthropod *Mollisonia symmetrica* from the mid-Cambrian (Wuliuan) Burgess Shale in British Columbia (Figs. 1–4 and Supplementary Figs. 1, 2). The Mollisoniida is an extinct group composed of only four valid genera (*Corcorania*, *Mollisonia*, *Thelxiope*, *Urokordia*) with occurrences in various Cambrian-Ordovician BST deposits of North America, South China, Wales, Morocco, and possibly Australia (reviewed in refs. [32,33]). Although mollisoniids are characterized by a distinctive exoskeletal morphology, including a head shield with deep anterolateral ocular notches, a parallel-sided thorax consisting of 7–14 freely articulating tergites, and a pygidial shield with marginal spines, until recently little had been known or even hypothesized concerning their relationships with other euarthropods[32]. The discovery of appendicular data in *Mollisonia plenovenatrix* from the Burgess Shale, suggesting a phylogenetic position within stem-group chelicerates, has prompted a substantial reevaluation of the significance of mollisoniids for euarthropod evolution[34]. We explore the significance of our palaeoneuroanatomical data on mollisoniids within the context of the chelicerate stem-lineage, and address its broader implications for the evolution of the euarthropod CNS organization by the mid-Cambrian.

## Results

**Neurological preservation in MCZ 1811**. MCZ 1811 is a completely articulated individual collected in 1925 by J. D. Houghton at the Raymond Quarry in Burgess Pass, and originally described by Percy Raymond in an upside-down orientation[35]. This specimen was designated as the holotype of the (now invalid) species

*Houghtonites gracilis* by P. Raymond[35], who also included within this new genus specimen USNM 57661 (Supplementary Fig. 3), the original holotype of *Mollisonia gracilis*[36]. Recent taxonomic revisions posit that *Houghtonites* represents a junior synonym of *Mollisonia*[32], particularly since the perceived differences between these taxa used for the original erection of a distinct genus were based on the erroneous orientation of MCZ 1811[35]. We follow the proposed synonymy of *Houghtonites* with *Mollisonia*, and formally assign MCZ 1811 to *M. symmetrica* (rather than *M. gracilis* as in ref. [34]) based on the similarity with other small-sized specimens that have been previously attributed to this species[32].

MCZ 1811 is preserved in dorsal view with a length (sag.) of 13 mm, and a maximum width (trans.) of 3.5 mm (Fig. 1). In accordance to the recently proposed diagnosis for Mollisoniida[32,33], the body outline is elongate, subrectangular, and maintains a consistent width throughout (parallel-sided). The exoskeleton consists of a head shield that represents approximately one-fifth of the body length (sag.), followed by a trunk composed of seven freely articulating tergites with lateral pleurae with subtriangular tips, and a pygidium (i.e., fused posterior shield) of similar dimensions to the head shield, and which bears two pairs of short marginal spines projecting from the posterior margin. The head shield is pentagonal in outline, and features a pair of anterolateral concave notches that accommodate well-developed, bulging lateral eyes. Between them, a pair of short, seemingly uniramous structures project anteriorly; their precise nature, especially whether they represent cephalic appendages, is uncertain owing to preservation (Fig. 1 and Supplementary Fig. 1). MCZ 1811 also shows traces of additional putative cephalic appendages in the form of a pair of delicate rami splayed at either side of the head shield, but their identity as either endopods or exopods remains inconclusive.

MCZ 1811 contains exceptional details of the non-exoskeletal anatomy. Each prominent lateral eye displays a light-colored outer cuticular layer that completely surrounds a dark inner region (Figs. 1, 2 and Supplementary Fig. 1). Backscattered electron microscopy and elemental mapping demonstrate that the inner dark regions contain substantial organic carbon films, and that they are directly connected to the head by carbonaceous threads (Figs. 1b, d and 2). The complex morphologies and relative positions of the light outer layer, the inner carbonaceous dark region, and the carbonaceous threads suggest that these features correspond to the eye lens, the pigmented retina, and the optic nerves of euarthropod lateral eyes, respectively[7]. The optic nerves of both eyes converge medially within the head shield, and continue posteriorly as a single, wider strand of organic carbon that corresponds to the cephalic CNS (Fig. 2 and Supplementary Fig. 1). However, the incomplete preservation of the cephalic CNS mass makes it impossible to distinguish discrete neurological structures, nor to confidently identify tracts innervating any head appendages. The rest of the body features a wide digestive tract that runs sagittally from the posterior two-thirds of the head to the anterior two-thirds of the pygidium (Fig. 1b). It is flanked laterally by large, apparently paired, ovoid to subrectangular structures that we interpret as paired gut diverticulae (Fig. 1b, d); the latter organs are rich in carbon, calcium, and phosphorous, which suggests that they have been phosphatized (Fig. 2), as frequently observed in non-biomineralizing euarthropods from Burgess Shale[9]. A similar composition is displayed by the gut tract, except in two regions where it is replaced by a material more comparable in composition to the matrix surrounding the fossil (i.e., rich in silicon and aluminum; Fig. 2), and therefore possibly representing gut sediment infill. Although most of the ventral nerve cord (VNC) is obscured by the overlying gut tract throughout the trunk, MCZ 1811 shows one possible segmental nerve in the pygidium. This structure is perpendicular to the

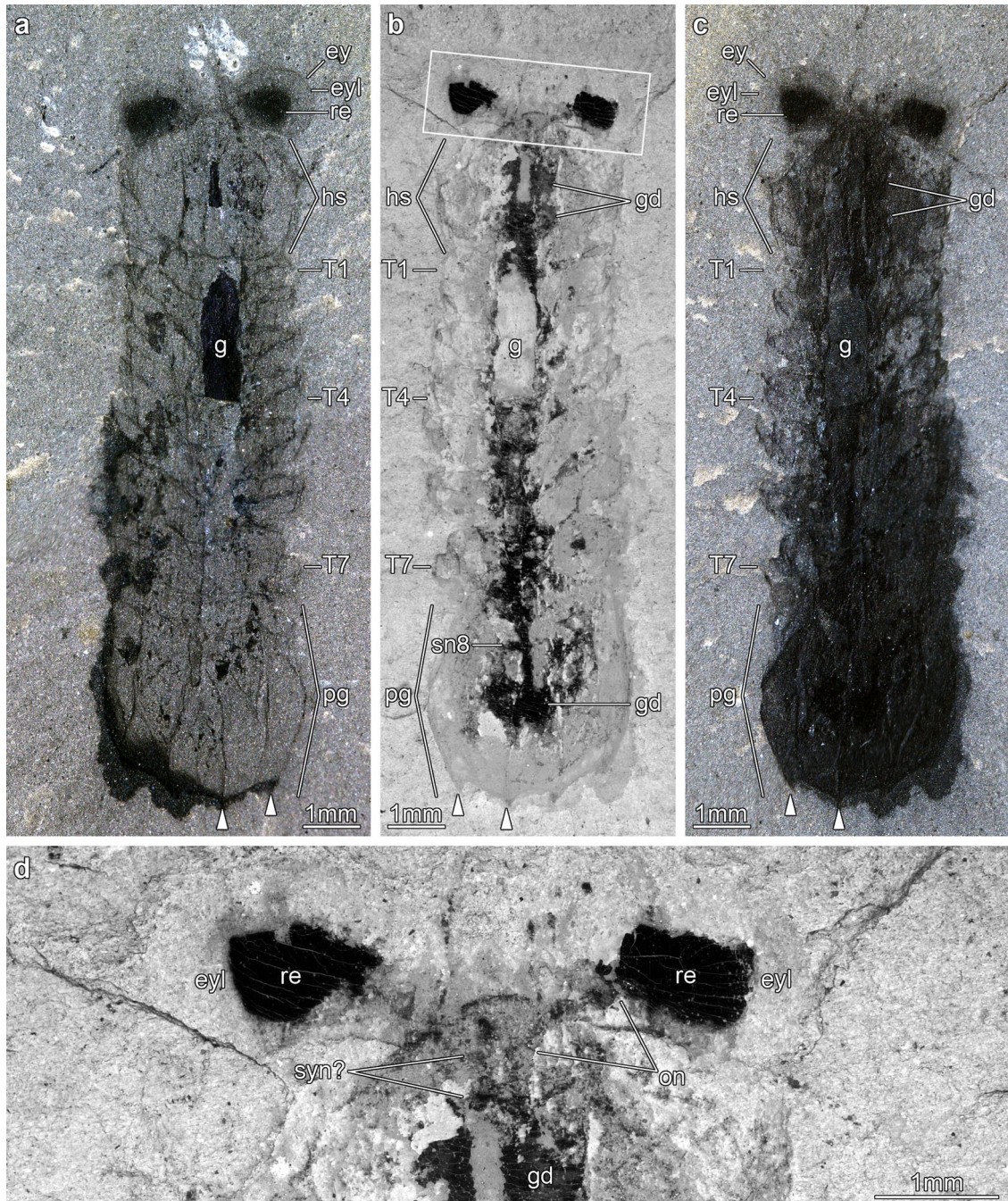

**Fig. 1 Central nervous system preservation in *Mollisonia symmetrica* from the Cambrian (Wuliuan) Burgess Shale, Raymond Quarry. a** MCZ 1811a, photographed underwater with cross-polarized illumination; arrowheads indicate position of spines in pygidium. **b** MCZ 1811b, imaged with backscattered electron microscopy. **c** MCZ 1811b, photographed underwater with cross-polarized illumination. **d** Magnification of head region in MCZ 1811b using backscattered electron microscopy. ey lateral eye, eyl eye lens, g gut tract, gd gut diverticulum, hs head shield, on optic nerve, pg pygidium, re retina, sn8 segmental nerve 8, syn? putative synganglion, T*n* trunk tergite.

digestive tract, but not directly connected to it, and is interpreted as possibly corresponding to the eighth pair of trunk appendages (Figs. 1b and 2). Alternatively, this structure could represent the incomplete remains of the digestive diverticula. The fossilized gut tract is split between part and counterpart of MCZ 1811. Close examination of the head region indicates that the anterior end of the gut tract is positioned on the layer above the cephalic CNS and corresponding optic nerves connected to the lateral eyes (Figs. 1d and 2). Elemental mapping reveals a significantly different composition of the cuticular exoskeleton (i.e., spatially variable, but consistently rich in silicon, and to a lesser extent

aluminum and oxygen), including the eye lenses, compatible with aluminosilicates (Fig. 2). The latter minerals likely result from the transformation of the original carbon film during the late metamorphic stage characterizing the taphonomic history of the Burgess Shale[5].

**Neurological preservation in USNM 305093.** USNM 305093 consists of a completely articulated individual (Fig. 3), which was collected by the Walcott family at the original Walcott Quarry that produced the holotype (USNM 57661; Supplementary Fig. 3)

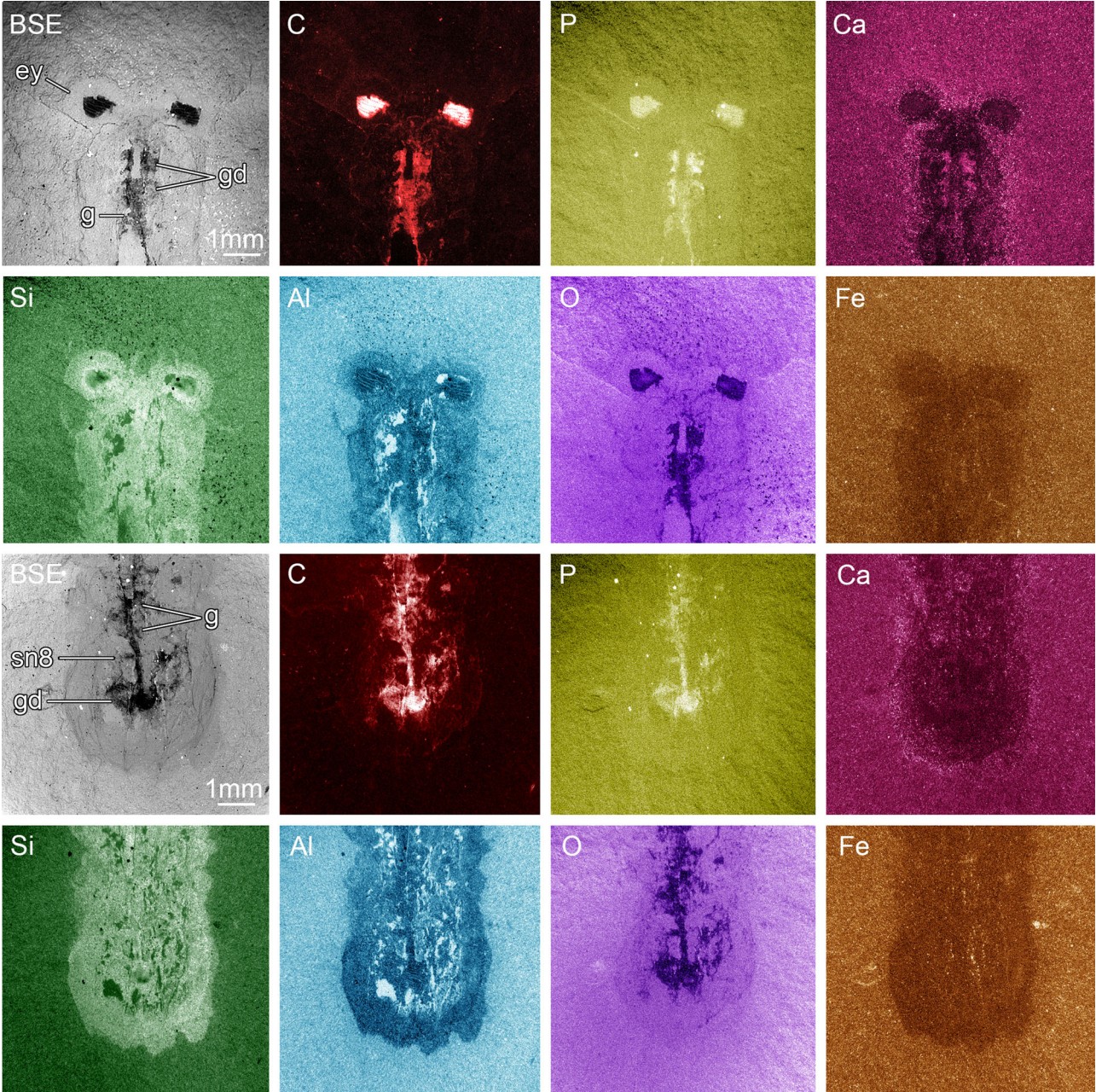

**Fig. 2 Elemental maps of MCZ 1811 head and posterior regions.** The specimen demonstrates preferential enrichment of carbon (C) and phosphorous (P) associated with labile soft tissues including the pigmented retina, optic nerves, gut tract, and paired diverticula. Furthermore, the presence of calcium (Ca) in the gut tract and diverticula indicates early diagenetic replacement by calcium phosphate. ey lateral eye, g gut tract, gd gut diverticulum, sn8 segmental nerve 8.

of "*Mollisonia gracilis*"[36], but not studied or figured until now. As-yet designated as *Mollisonia* sp. in the Smithsonian Institution Invertebrate Paleontology collections, we interpret USNM 305093 as a juvenile individual of *Mollisonia symmetrica* based on its small size, elongate general morphology, lack of genal spines, and the presence of three pairs of marginal spines on the pygidium. *M. symmetrica* is also the sole valid species of *Mollisonia* reported from this locality to date[32].

USNM 305093 is preserved in lateral view, with a length (sag.) of 7.5 mm and a maximum height (trans.) of 1.7 mm (Fig. 3). As with all members of Mollisoniidae[32], the dorsal exoskeleton consists of a head shield, a trunk with seven freely articulating tergites with well-developed pleurae, and a pygidium with posteriorly facing marginal spines. USNM 305093 lacks traces

of preserved appendages, with the exception of a possible limb ramus below the head shield (Fig. 3c and Supplementary Fig. 2). Carbonaceous structures are not readily apparent under reflected light (Fig. 3a, b), but cross-polarized photography of the specimen under wet contrast reveals exceptional details of the neuroanatomy (Fig. 3c). Despite being clearly visible, backscattered electron imaging and elemental mapping did not detect substantial carbonaceous films in the internal anatomy of USNM 305093. As in MCZ 1811 (Figs. 1, 2 and Supplementary Fig. 1), a pair of bulbous lateral eyes protrude in front of the head shield, and each of these eyes is composed of a light-colored outer lens surrounding a discoid, dark-colored (Fig. 3b) or highly reflective (Fig. 3c and Supplementary Fig. 2) pigmented retina. Thick threads, interpreted as the optic nerves, run posteriorly from the

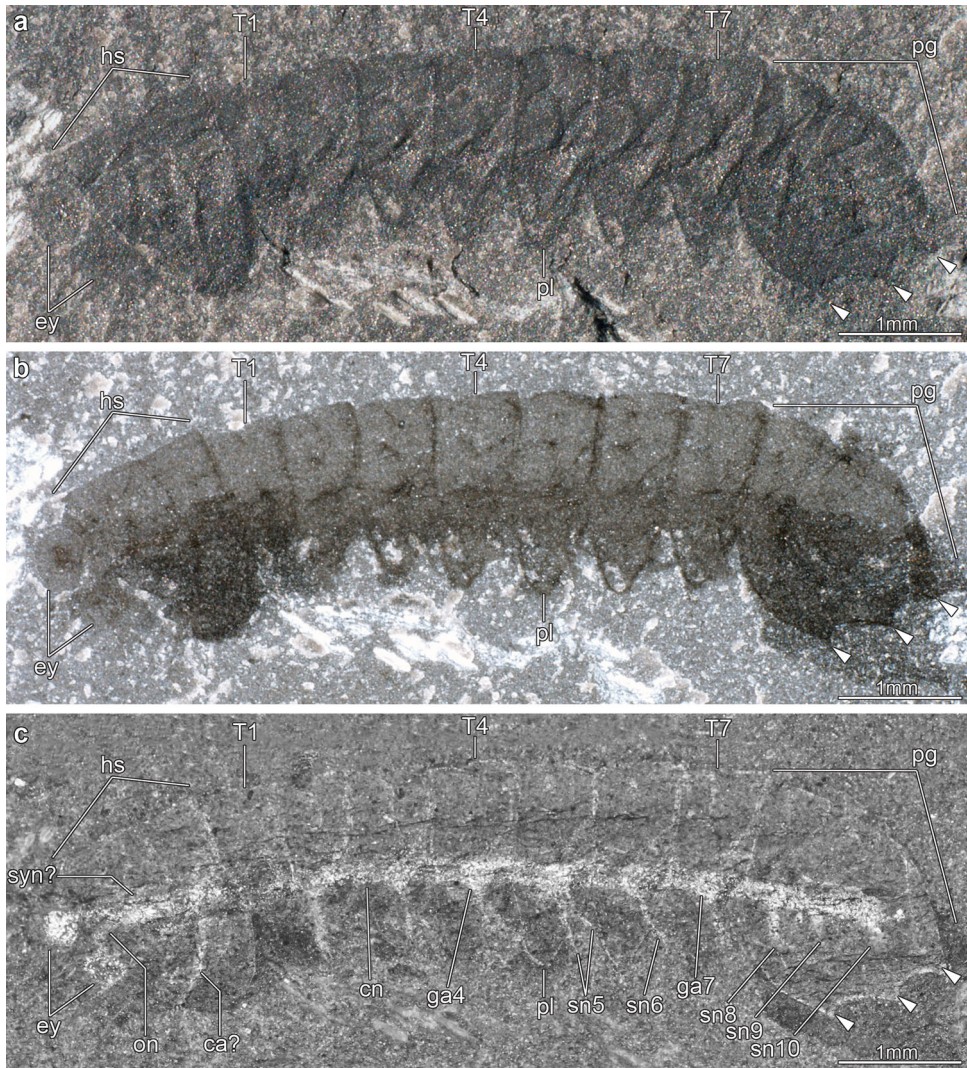

**Fig. 3 Central nervous system preservation in *Mollisonia symmetrica* from the Cambrian (Wuliuan) Burgess Shale, Walcott Quarry. a** USNM 305093, photographed dry with low angle reflected light to enhance topographic information; arrowheads indicate the position of spines in the pygidium. **b** USNM 305093, photographed dry with cross-polarized illumination to enhance specimen outline. **c** USNM 305093, photographed underwater with cross-polarized illumination to highlight morphology of central nervous system. ca? putative cephalic appendage, cn connective, ey lateral eye, ga*n* segmental ganglion, hs head shield, on optic nerve, pl pleura, pg pygidium, sn*n* segmental nerve, syn? putative synganglion, T*n* trunk tergite.

eyes toward the midline, where they merge into a single axial strand of highly reflective material that extends throughout the entire body. This axial strand is regarded as the CNS. Its cephalic portion is uniform in width (ca. 130 μm, trans.) and degree of reflectivity, except for a slightly wider posterior region which gives rise to a possible lateral nerve or cephalic appendage running toward the head shield left margin (Fig. 3c and Supplementary Fig. 2). Beyond its overall outline, the cephalic CNS lacks recognizable neuroanatomical features, such as distinct neuromeres, segmental nerves, or the esophageal foramen. By contrast, the VNC in the thoracic region is well defined, and consists of condensed ganglia, up to ca. 160 μm-wide (trans.), which are linked together by slender longitudinal connectives (Fig. 3c and Supplementary Fig. 2). The ganglia are metamerically organized, with each lying medially under each of the seven thoracic tergites. Several ganglia feature the proximal remains of ventrolaterally projecting bifurcating segmental nerves that would innervate the ventral limbs. These segmental nerves occasionally extend beyond the lateral margins of the tergites (e.g., the fifth and sixth segmental nerves; Figs. 3c and 4a). There are no well-differentiated ganglia under the pygidium, but three sets of

conspicuous segmental nerves are visible, despite the absence of freely articulating tergites. Along with the three pairs of marginal spines (Fig. 3c and Supplementary Fig. 2) and dorsal ridges (the latter are not visible in this small specimen, but see Supplementary Fig. 3), these nerves support the hypothesis of a three-segmented pygidium in *Mollisonia*[33]. The axial reflective strand can be confidently interpreted as the CNS based on the direct connection to the eyes, discretely metameric and bilaterally symmetric organization, and detailed morphology including condensed ganglia with laterally oriented segmental nerves. An in-depth discussion of the taphonomy of mollisoniid neurological tissues is provided in the Supplementary Information.

## Discussion
**Contrasting appendicular, exoskeletal, and neurological data.** MCZ 1811 and USNM 305093 offer complementary dorsal and lateral views (respectively) that allow reconstructing the CNS of *Mollisonia symmetrica* almost in its entirety (Fig. 4). A closer comparison with the exoskeletal and appendicular data of *Mollisonia plenovenatrix*[34] further informs the correlation between

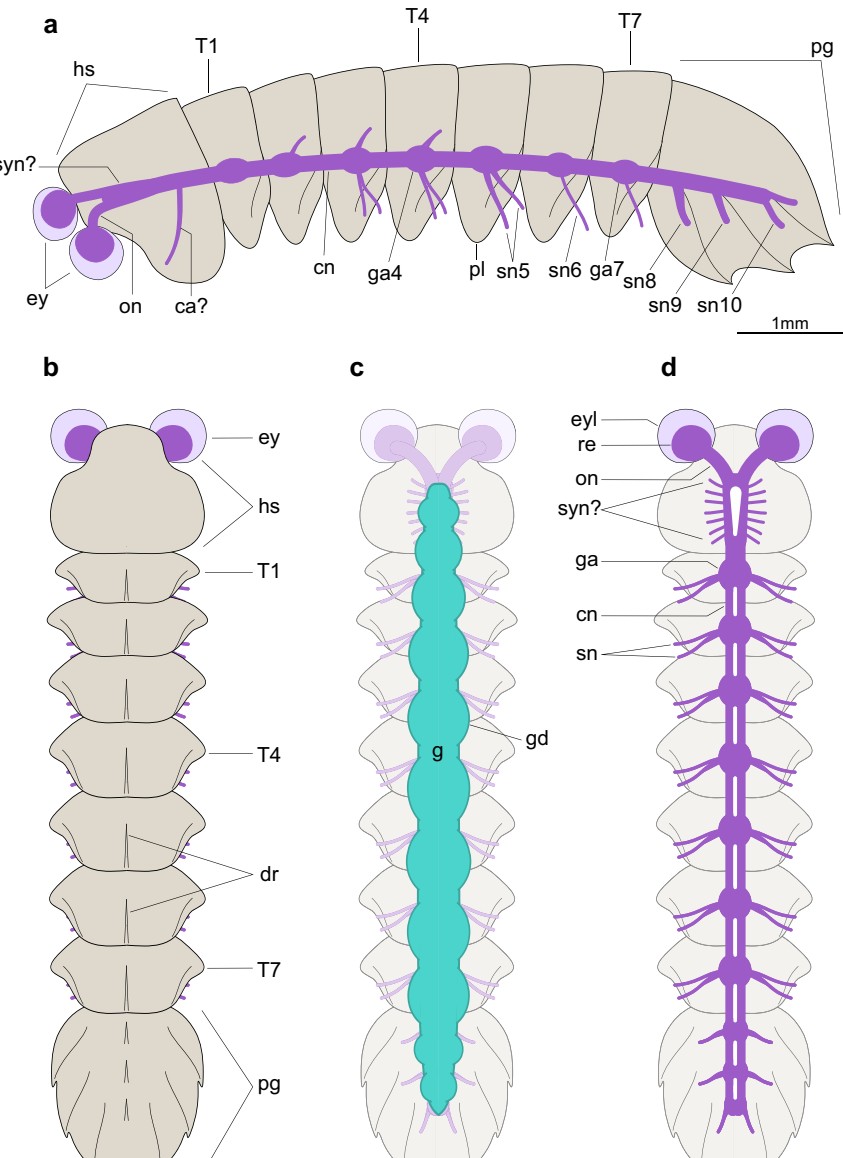

**Fig. 4 Morphological interpretation and reconstruction of mollisoniid exoskeletal and internal anatomy. a** USNM 305093, interpretative diagram.
**b** Dorsal morphology of *Mollisonia symmetrica*; appendicular data is omitted due to its incompleteness. **c** Internal anatomy showing the digestive system.
**d** Central nervous system organization. The morphology and number of segmental nerves in the putative synganglion and esophageal foramen within the
head shield are hypothetical, based on the appendicular data described for *Mollisonia plenovenatrix* from Burgess Shale and its suggested affinities with
chelicerates[34]. ca? putative cephalic appendage, cn connective, dr dorsal ridge, ey lateral eye, eyl eye lens, g gut tract, ga*n* segmental ganglion, gd gut
diverticulum, hs head shield, on optic nerve, pl pleura, pg pygidium, re retina, sn*n n segmental nervesegmental nerve,* syn? putative synganglion, sn*n*
segmental nerve, T*n* trunk tergite.

internal and external anatomy. *M.plenovenatrix* was described as
having a pair of well-developed lateral eyes, seven pairs of
cephalic limbs (including putative minute chelicerae, three uni-
ramous appendages, and three possibly biramous appendages),
and 11 pairs of trunk appendages (seven thoracic, four pygidial)
consisting of oval-shaped flattened exopods[34]. The prominent
lateral eyes of *M. plenovenatrix* are also expressed in *M. sym-
metrica*, albeit with a less oblong shape, and our studied speci-
mens clearly show how the optic nerves converge towards the
median region of the head (Figs. 1d, 2, 3c and Supplementary
Fig. 1, 2). Although the prominent eyes of *M. plenovenatrix* have
been described as having three condensed dark masses inter-
preted as nested optic neuropils[34], we find no evidence of such
structures in *M. symmetrica* (Figs. 1, 3 and Supplementary Fig. 1).

In particular, the previous claim of up to two optic neuropils
composing the dark inner region of the lateral eyes in MCZ
1811[34] could not be confirmed by direct study of the specimen
using optical and backscattered electron microscopes (Fig. 1 and
Supplementary Fig. 1). The one-to-one correlation between the
thoracic tergites and corresponding exopods in *M. plenovenatrix*
perfectly matches the disposition of the VNC ganglia observed in
USNM 305093 (Fig. 3c and Supplementary Fig. 3). However, the
occurrence of three segmental nerve bundles in the pygidium of
USNM 305093 (Fig. 3c) confirms that the pygidium of molliso-
niids likely includes three, rather than four non-articulated
appendage-bearing segments (*contra* ref. [34] see discussion in
ref. [33]). Additional evidence for this proposal comes from the
presence of three pairs of dorsal ridges and marginal spines in the

pygidium, which are regarded as serial homologs of the pleural ridges and tips, respectively, of the thoracic tergites[33]. These observations indicate that the mollisoniid trunk follows the ancestral one-to-one correspondence between segmental ganglia and trunk tergites (e.g., ref.[37]), rather than showing indications of substantial mismatch between the CNS and the dorsal exoskeleton as observed in some Cambrian euarthropods[8,19].

The head region shows a greater discrepancy between the appendicular anatomy and expected CNS complexity. Whereas *M. plenovenatrix* has been interpreted as having seven pairs of appendages closely packed within the head shield[34], the preserved CNS in USNM 305093 lacks evidence of substantial differentiation (Fig. 3c). The absence of distinguishable neurological characters (e.g., brain neuromeres, segmental nerves, esophageal foramen) in the cephalic CNS of USNM 305093 can be explained in two ways. Incomplete preservation of the cephalic CNS could be responsible for the lack of clear morphological data. Alternatively, it is also possible that the cephalic CNS reflects a tight functional integration of the seven cephalic limb pairs in *Mollisonia* (based on *M. plenovenatrix*) expressed as a highly condensed synganglion-like structure, similar to that expressed in the prosoma of extant euchelicerates, but too compact to be discernible in such a small specimen through BST preservation. For context, Cambrian euarthropod specimens exhibiting conspicuous cephalic nerves are at least three times larger (e.g., *Alalcomenaeus* sp. is ca. 22–60 mm in body length[16,22]), and preserve significantly bigger brains (e.g., ca. 2 mm in width in *Fuxianhuia protensa*[28] and in *Odaraia alata*[18]) when compared to USNM 305093 (7.5 mm body length, and ca. 130 μm brain width). There is no clear evidence for encephalization in our studied specimens of *Mollisonia*, but future discoveries of neuroanatomical structures in larger specimens will most likely inform the CNS of the head region in greater detail. Despite these limitations, the fact that the organization of the CNS closely parallels that of the body (as expressed by the dorsal exoskeleton) suggests that our data reliably capture the general neuroanatomy of these Cambrian euarthropods (Fig. 4).

**Mollisoniids reveal the ancestral condition of chelicerate CNS.** Mollisoniids have been historically regarded as problematic owing to the scarcity of limb preservation and lack of clear synapomorphies that would link them to other groups of Cambrian euarthropods[32,33,38], but recent phylogenetic analyses have investigated their affinities. A large data set of total-group Euarthropoda recovered mollisoniids as an early branching monophyletic clade of trilobitomorph artiopods within stem-group Chelicerata[39]. Incorporating appendicular data on *Mollisonia plenovenatrix* (e.g., minute chelicerae, trunk book gill-like exopods), a more recent analysis with smaller taxon sampling resolved *Mollisonia* as a stem-group chelicerate and the antennae-bearing artiopods as stem-group Mandibulata[34]. In order to explore the broader implications of *M. symmetrica*'s palaeo-neuroanatomical data in the context of early chelicerate nervous system evolution, we performed Bayesian inference and maximum parsimony phylogenetic analyses (including equal and implied weights) that incorporate all known Cambrian taxa with neurological preservation (Supplementary Note; Supplementary Data 1). Mollisoniids are recovered as sister to a clade uniting megacheirans and euchelicerates, and therefore as the earliest diverging member of stem-group Chelicerata in all analyses (Supplementary Fig. 5).

*Mollisonia* reveals a mosaic of ancestral and derived neurological characters relative to other members of total-group Chelicerata (Fig. 5). *Mollisonia*'s metamerically arranged VNC with regularly spaced condensed ganglia is also expressed in the

trunk of extant pycnogonids[40,41], xiphosurids[42,43], and arachnids[44]. These comparisons suggest that *Mollisonia*'s VNC embodies the ancestral organization for total-group Chelicerata (Fig. 5a), and could possibly even reflect that of crown-group Euarthropoda given the early branching position of mollisoniids within the chelicerate stem lineage[34,39] (Supplementary Fig. 5). *Mollisonia*'s lateral eyes and their connection with the dorsal brain also suggest a plesiomorphic organization of the CNS compared to crown-wards representatives. Whereas several extant chelicerates (e.g., *Limulus*, arachnids) share the presence of paired sets of eyes (or doublets), each containing a single neuropil that lies outside of the protocerebrum[16], *Mollisonia* features a single set of lateral eyes (Figs. 1–4). The presence of single lateral eyes in *Mollisonia* likely illustrates the ancestral organization of the chelicerate stem lineage before the evolution of the derived ocular and neurological characters found in extant representatives. Furthermore, the phylogenetic position of *Mollisonia* within stem-group Chelicerata (*sensu* refs. [34,39]) (Fig. 5a and Supplementary Fig. 5) suggests that aspects of *Mollisonia*'s neuroanatomy may reflect the ancestral organization of crown-group Euarthropoda (Fig. 5a), as for instance the homonomous VNC in the trunk that is also known in mandibulates[45–47]. *Mollisonia* also confirms that the evolutionary reduction in the number of intersegmental peripheral nerves in the VNC represents a synapomorphy of crown-group Euarthropoda, which contrasts with the scalidophoran-like ancestral VNC organization observed in the fuxianhuiid (stem-group Euarthropoda) *Chengjiangocaris kunmingensis*[19].

**Mollisoniid CNS support chelicerate affinities of megacheirans.** Our results carry direct implications for understanding the diversity of the chelicerate stem group and the early evolution of the CNS in this lineage. In addition to mollisoniids, other proposed members of the Cambrian stem-group of Chelicerata include *Habelia*[48], sanctacaridids[49,50], and megacheirans[51,52]. Whereas there is growing support for the chelicerate affinities of *Habelia* and sanctacaridids based on their appendicular organization, and that both groups are close relatives of mollisoniids, the position of megacheirans has remained controversial[31,53,54]. Megacheirans are non-biomineralized euarthropods typified by the presence of a well-developed pair of raptorial "great appendages", which have been interpreted as precursors of the chelicerate chelicerae (e.g., refs. [16,51,52,55] but see ref. [56] for an alternative hypothesis). They also share with euchelicerates the presence of a reduced labrum[57,58], as recently documented in *Leanchoilia illecebrosa*[55], and several neuroanatomical features. Within Megacheira, neuroanatomical data is known from the leanchoiliids *Alalcomenaeus* and *Leanchoilia*, with the CNS organization in these taxa corroborated by multiple specimens from the lower and middle Cambrian of South China[16,24] and North America[22] that are preserved in dorsal and lateral views. The CNS of *Alalcomenaeus* supports a deutocerebral segmental origin for megacheiran great appendages, and also shows derived neurological characters shared with extant euchelicerates including the lateral eye doublets with a single neuropil outside the protocerebrum, a condensed synganglion-like structure that innervates the cephalic appendages, an extended esophageal foramen, and the presence of multiple connectives that extend into the posterior end of the body (Supplementary Fig. 1)[7,16,22]. The results of our phylogenetic analysis indicate that palaeoneuroanatomical data support a crown-ward position of megacheirans relative to mollisoniids within the chelicerate stem lineage, and thus a step-wise evolution of the CNS in this group (Fig. 5a and Supplementary Fig. 5). Although the presence of seven pairs of tightly packed appendages in the head of *M. plenovenatrix*[34] suggests the presence of a condensed synganglion, confirmation of this feature will

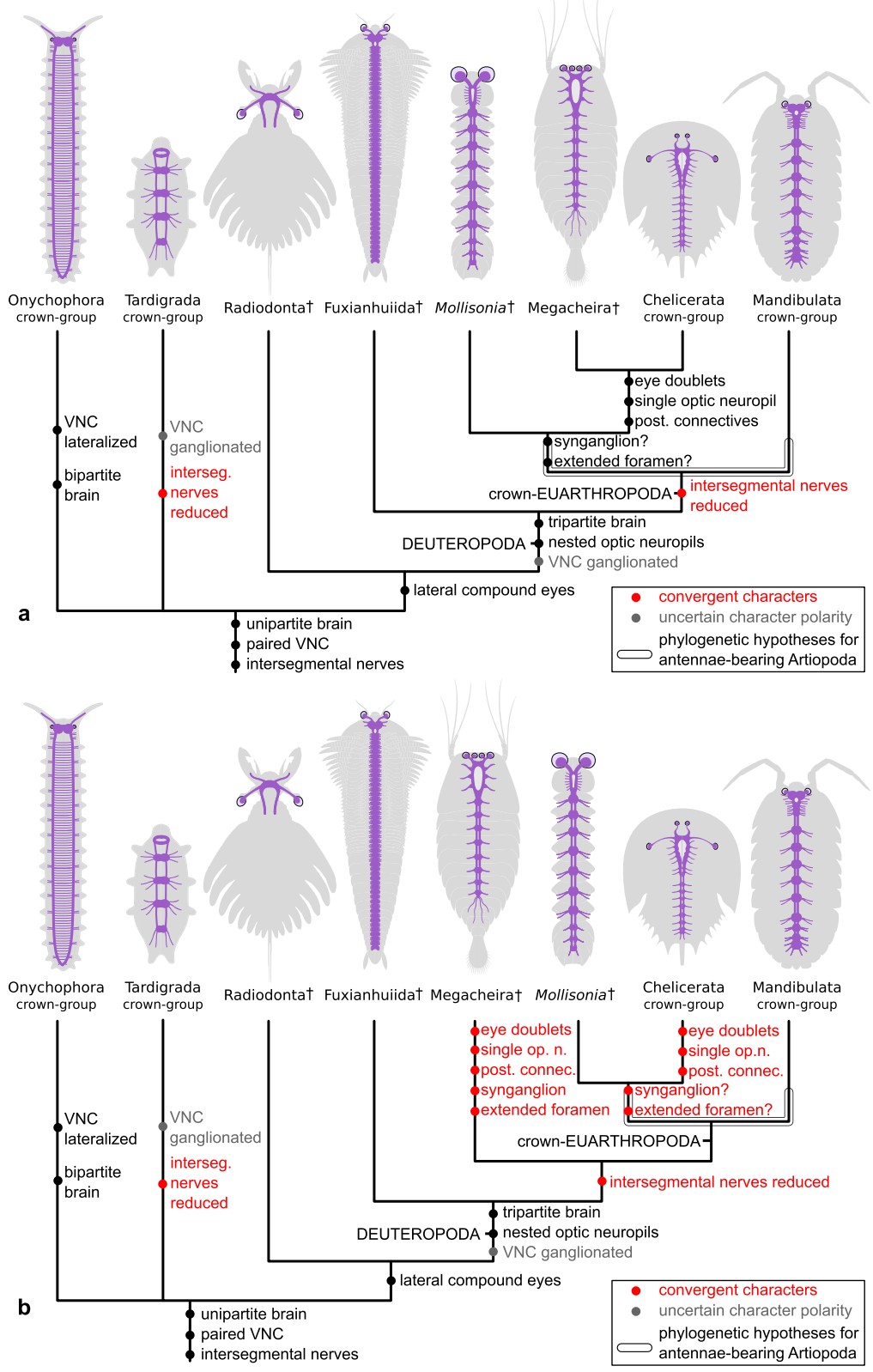

require additional mollisoniid fossils with good anterior CNS preservation.

A competing hypothesis, supported by several phylogenetic analyses[34,39,56], resolves megacheirans as a grade of stem-group euarthropods (Fig. 5b). However, such an early branching position implies the convergent evolution of multiple neuroanatomical and appendicular characters shared between megacherians

and euchelicerates[59], including the eye doublets with a single optic neuropil outside the protocerebrum, the condensed cephalic synganglion with extended esophageal foramen, the presence of posterior longitudinal connectives in the VNC, the raptorial first pair of appendages, and a reduced labrum[16,22,55]. A position of megacheirans within total-group Chelicerata is more parsimonious (Supplementary Fig. 5), as it involves a single origin for several

**Fig. 5 Evolution of the central nervous system in Panarthropoda.** Known neuroanatomical organization depicted in purple. Crosses indicate extinct clades. **a** Simplified tree topology showing the position of *Mollisonia* and Megacheira as members of stem-group Chelicerata, based on the results of the phylogenetic analyses under both Bayesian inference and maximum parsimony (Supplementary Fig. 5); this hypothesis produces the most parsimonious reconstruction of the evolution of the central nervous system. **b** Alternative tree topology showing the position of *Mollisonia* within stem-group Chelicerata, and megacheirans as stem-group Euarthropoda (e.g., refs. [34, 43]) this topology implies convergent evolution of the central nervous system between megacheirans and crown-group chelicerates, and was not recovered in our phylogenetic analyses. Sources for neuroanatomical data: Onychophora[71], Tardigrada[72], Radiodonta[17], Fuxianhuiida[15, 19], Megacheira[16, 22], Chelicerata[43], *Mollisonia* (this study), Mandibulata[47]. Character coding is available in the Supplementary Note. Morphological data set of 54 taxa and 106 characters available in the Supplementary Data 1. post. connec. posterior connectives, single op. n. single optic neuropil outside protocerebrum.

morphological characters shared between megacheirans and crown-group chelicerates (Fig. 5a). It also finds additional support from the several external morphological characters shared between megacheirans, *Habelia*, and sanctacaridids, such as ventral stalked eyes (expressed in all of them), the presence of paired axial protrusions on the tergites (e.g., *Habelia*, *Sanctacaris*, *Wisangocaris*, *Leanchoilia*) and a paddle-shaped tailspine with marginal spines (e.g., *Sanctacaris*, *Utahcaris*, *Wisangocaris*, and *Alalcomenaeus*, *Leanchoilia*, *Yohoia*). Regardless of the position of megacheirans, both evolutionary scenarios carry the implication that the CNS of Pycnogonida[40,41] appears to have undergone substantial secondary simplification relative to Cambrian stem-group representatives as informed by *Mollisonia* and *Alalcomenaeus*, whilst still retaining a distinctly ganglionated VNC organization that closely resembles that of *M. symmetrica* (Fig. 3).

**Mozaic evolution in the chelicerate stem-lineage.** The description of the CNS of *Mollisonia* reveals that appendicular and neuroanatomical data may carry conflicting phylogenetic signals in stem-group chelicerates. Whereas neurological characters strongly suggest a more derived condition in megacheirans compared to mollisoniids, the opposite is true for appendicular data. The presence of small chelicerae, possibly up to seven pairs of cephalic appendages, and book gill-like trunk exopods in *Mollisonia plenovenatrix*[34] would favor a more crown-ward position relative to megacheirans, which feature deutocerebral great appendages, four pairs of cephalic appendages, and biramous trunk appendages (Fig. 5a). The lack of direct phylogenetic congruence between appendicular and neurological data denotes a degree of evolutionary independence of these two aspects of the anatomy in early chelicerates. This discrepancy recalls the presence of a mandibulate-like tripartite brain and lateral eyes with multiple nested neuropils in *Fuxianhuia protensa*[15,28], despite the fact that the fuxianhuiid VNC features an ancestral scalidophoran-like organization[19], and that fuxianhuiids do not share derived appendicular or exoskeletal characters with known members of stem-group Mandibulata. Thus, the occurrence of mosaic evolution in the external and internal anatomy of the earliest chelicerates would not be unexpected, considering that this evolutionary pattern has been documented in various groups (e.g., refs. [60–62], including humans[63]), and regarded as pervasive in the history of the biosphere[64]. Although the Cambrian fossil record of stem-group chelicerates remains too incomplete to properly test the rates of evolutionary change between the external and internal anatomy, future discoveries will continue providing additional data to test this hypothesis. In the meantime, our palaeoneurological data from *Mollisonia symmetrica* provides a valuable glimpse into the overall ground pattern for the CNS of crown-group Euarthropoda based on mid-Cambrian representatives.

## Methods

**Fossil imaging and elemental analysis.** Specimens were photographed with a Nikon D850 DSLR fitted with an AF-S Micro Nikkor 60 mm lens (MCZ), and an Olympus DSX-110 inverted digital microscope (USNM). Backscattered scanning electron microscopy and energy dispersive spectroscopy-based elemental mapping were performed with a QEMSCAN 650 F (10 kV, WD: 14.6 mm) at the Department of Earth Sciences at the University of Cambridge, and a custom Tescan Vega GMU (20 kV, WD: 15 mm) at Harvard University. Figures were produced in Photoshop CS and Inkscape.

**Phylogenetic analysis.** The character matrix used for the phylogenetic analyses includes 54 taxa and 106 characters (Supplementary Data 1). We employed an updated version of the data set used by Yang et al.[19], which is specifically designed to incorporate palaeoneuroanatomical data from Cambrian euarthropods. Character coding and discussion of results are available in the Supplementary Note.

The Bayesian analysis was run in MrBayes 3.2 using the Monte Carlo Markov-chain model for discrete morphological characters[65,66] for 5 million generations (four chains), with every 1000th sample stored (resulting in 5000 samples), and 25% burn-in (resulting in 4000 retained samples). Convergence was diagnosed with the software Tracer[67], with effective sample size values over 200. The parsimony analyses were run in TNT[68] under New Technology Search, using Driven Search with Sectorial Search, Ratchet, Drift, and Tree fusing options activated with standard settings[69,70]. The analysis was set to find the minimum tree length 100 times and to collapse trees after each search. All characters were treated as unordered. For comparative purposes, analyses were performed under equal and implied weights ($k = 3$).

**Reporting summary.** Further information on research design is available in the Nature Research Reporting Summary linked to this article.

## Data availability

The studied fossil specimens are deposited at the Harvard University Museum of Comparative Zoology in Cambridge (MCZ) and the Smithsonian Institution in Washington D. C. (USNM). No new fossil specimens were collected for this study. No permissions were required for performing this research. Character coding for the phylogenetic analysis is available in the Supplementary Information. The morphological data set is available in Supplementary Data 1 in both text and excel formats.

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

## Acknowledgements

Thanks to Jessica Cundiff (Museum of Comparative Zoology, Harvard University), Mark Florence, and Douglas Erwin (Smithsonian Institution) for facilitating access to Burgess Shale collections, and Scott Whittaker (Smithsonian Institution) for training and

facilitating access to imaging facilities. Thanks to Iris Buisman (University of Cambridge) for assistance with use of electron microscopy facilities for preliminary elemental analyses at the Department of Earth Sciences, University of Cambridge (UK). Nicholas Strausfeld (University of Arizona) kindly shared photographs of the megacheiran *Alalcomenaeus*. Published by a grant from the Wetmore Colles Fund.

## Author contributions

J.O.-H. designed research, photographed fossil material, prepared figures, and performed phylogenetic analysis. J.O.-H and R.L.-A. interpreted results and wrote the manuscript with input from all authors. S.R.L. photographed fossil material, obtained elemental data, and prepared figures. J.C.W. obtained elemental data. All authors discussed and approved the final manuscript.

## Competing interests

The authors declare no competing interests.
