## [Peer Review File · Nature Communications]

Neuroanatomy in a middle Cambrian mollisoniid and the ancestral nervous system organization of cheliceratesREVIEWER COMMENTS

Reviewer #1 (Remarks to the Author):

Fossilized neural structures, particularly in Cambrian exceptionally preserved fossils, have been becoming pervasive in last decades, although arguments from taphonomic perspective urge cautious interpretations. Here Ortega-Hernandez et al. report a further case, the nearly complete neural system of mollisoniid arthropods from the middle Cambrian of Northern America. In terms of current criteria to identify fossil neural structures, I agree that these fossils do preserve potential neuroanatomy of mollisoniids.

I have some concerns need the authors to help clarify though.

1) as the authors described, the preservation of mollisoniid nerve is not complete, some interpreted anatomic features only represent one of the possible interpretations (e.g. synganglia). In this sense, I would hesitate to accept the discussions concerning the 'significance' of this discoveries. The discussion part is too long, most of which are of possibilities, not of the discovery.

2) In Fig 2, element P and Ca show about four paired lateral extensions. Are those true signals ?

3) In MCZ 1811, the interpreted CNS is quite a thin thread in the pygidium, while in USNM 305093, the interpreted CNS is quite thick and robust. This would indicate that the later CNS is more likely the combination of gut and CNS, like the central carbon remains in MCZ 1811.

4) It is really hard to believe that the girth of mollisoniids can reach the width as indicated in Fig. 4C. It looks more like the body width under the tergites. Supporting data are needed for such a claim.

5) In Fig 2. Interpretative labelings are needed to help understand the authors' claims

6) As the authors notified, mollisoniids are widely accepted as stem chelicerates. I am not sure whether it is right to exaggerate the significance to euarthropods, given that some detailed neuroanatomy are not that complete and informative.

In summary, this is an additional interesting case of fossil nerves in Cambrian arthropods. The findings will further motivate the discussion about the early evolution of arthropods, from the perspective of neuroanatomy. I would recommend to publish this paper if the authors can address the questions raised above.

Peiyun Cong
YKLP, Yunnan University

Reviewer #2 (Remarks to the Author):

Neural structures are rarely preserved in the fossil record, so despite the increasing reports of exceptionally preserved neuroanatomy from Cambrian panarthropod fossils in the past few years, there is still a huge amount we don't know. The manuscript by Ortega-Hernández and his co-authors described the central nervous system (CNS) of the Cambrian arthropod *Mollisonia symmetrica* from the Burgess Shale. The authors carried out careful observations, detailed descriptions, and the manuscript is well-written and well-illustrated. Therefore, this study is certainly a welcome addition to the new research field of "neuropalaeontology", and the evidence and discussion in the manuscript is generally robust and sound. However, I have a few concerns.

1) I have my reservations regarding the optical neuropils of *Mollisonia*. The number and position of

optical neuropils are very important characters for distinguishing mandibulate-like or chelicerate-like brains. In my opinion, the evidence from ROMIP 62978 in Ref [32] convincingly shows that *Mollisonia* has three optic neuropils nested outside the central brain, and the outline of two optic neuropils can also be recognised in specimen MCZ 1811 (Extended Data Fig. 5 in Ref [32]; and Fig. 1 and S1 in this manuscript), so such evidence shouldn't be dismissed easily. Therefore, the single dark area of preservation within the eye stalk in other specimens might be the result of an overlaying of retina and optic neuropils.

2)The arguments in the manuscript are somehow circular or even conflicting. The author cautiously emphasized several times that the poor preservation of the anterior CNS of *Mollisonia* causes uncertainty regarding the detailed features of the CNS (e.g. optic neuropil discussion above). If so, the discussion about the phylogeny based on the CNS will be built on a weak foundation, and hence there is no real evidence of conflicting phylogenetic signals between appendicular or neuroanatomical data. For the same reason, it is not clear about the what is being illuminated with regards to the ancestral nervous system of crown-group Euarthropoda. The author can change the title and tone down some statements in the Discussion.

3)The phylogenetic position of *Mollisonia* has been controversial among different previous studies, which are all based on appendicular data. Neuroanatomic characters are often very useful for solving deep phylogeny. Therefore, I wonder if the authors can carry out new phylogenetic analysis with the addition of new neural characters to address this controversy? If so, some of my suggestions above would be solved too.

A few minor points below:

1)There is some repetition in the manuscript, such as preservation in the Results and in the Discussion, which could be consolidated.

2)Line 405: (Figure S5)--There is no figure.

3)In Fig. 3 caption: "ssn" should be "sn"

4)Fig. S2 is identical to Fig. 3, apart from the figure caption

REVIEWER COMMENTS

Reviewer #1 (Remarks to the Author):

Fossilized neural structures, particularly in Cambrian exceptionally preserved fossils, have been becoming pervasive in last decades, although arguments from taphonomic perspective urge cautious interpretations. Here Ortega-Hernandez et al. report a further case, the nearly complete neural system of mollisoniid arthropods from the middle Cambrian of Northern America. In terms of current criteria to identify fossil neural structures, I agree that these fossils do preserve potential neuroanatomy of mollisoniids.

JOH: We thank the reviewer for the positive feedback.

I have some concerns need the authors to help clarify though.

1) as the authors described, the preservation of mollisoniid nerve is not complete, some interpreted anatomic features only represent one of the possible interpretations (e.g. synganglia). In this sense, I would hesitate to accept the discussions concerning the 'significance' of this discoveries. The discussion part is too long, most of which are of possibilities, not of the discovery.

JOH: The reviewer is correct in the presence of the synganglion being one of the two aspects of the nervous system that is unclear in *Mollisonia symmetrica*. The other is the presence of recognizable neuropils, but these structures are well-described in *M. plenovenatrix*. The specific issue of the presence/absence of a synganglion should not be overestimated, and definitely not minimize the significance of the less questionable (and undisputed) findings of our study, which concern more important aspects of the neurological organization of *Mollisonia symmetrica*. The presence of single eyes, and a clearly ganglionated ventral nerve cord with segmental nerves differs markedly from the neuroanatomy of extant chelicerates (e.g. *Limulus*) and other members of the chelicerate stem lineage, particularly the megacheirans (see Tanaka et al. 2013). Importantly, the absence of a synganglion would only provide further support to our claim that mollisoniid CNS documents the most plesiomorphic condition within total-group chelicerates.

To strengthen our conclusions, we have included a phylogenetic analysis that is specifically designed to evaluate the implications of palaeoneuroanatomical data in Cambrian fossils. This lends support to our claim that *Mollisonia* most likely reflects a plesiomorphic condition of the chelicerate nervous system. We have modified the discussion accordingly to incorporate the results of the analyses, and provided the full results in Supplementary Figure 5. We hope that the present explanations and the changes made to our manuscript will convince referee 1 that our discussion is first and foremost based on confident new observations, while uncertain aspects of mollisoniid anatomy were treated in the most conservative way.

As to the excessive length of the discussion, we have moved the first section focusing on preservation to the supplementary information, considerably shortened the other two sections, and deleted the paragraph on the possible implication for our understanding of euarthropod CNS (see referee 1's concern #6 below).

2) In Fig 2, element P and Ca show about four paired lateral extentions. Are those true signals?

JOH: That is correct. The enrichment in Ca and P in the axial region is restricted to structures that we interpret as the gut wall and gut diverticulae, based on their location and shapes. As mentioned in our introduction, authigenic phosphatization is a particular common mode of preservation for euarthropod gut structures in Burgess Shale-type deposits (e.g Butterfield 2002; Lerosey-Aubril et al. 2012; Vannier et al. 2014). We have included additional labels to Figure 2 to illustrate our interpretation of these structures as clearly as in Figure 1.

3) In MCZ 1811, the interpreted CNS is quite a thin thread in the pygidium, while in USNM 305093, the interpreted CNS is quite thick and robust.

JOH: To clarify, we argue that the preservation of the CNS in the pygidial region of MCZ 1811 is restricted to a possible segmental nerve (sn8?). Other axial traces in this area more likely correspond to remains of the digestive system. This includes the thin sagittal trace that seems to be misinterpreted by referee one as the VNC- close examination shows that it is the posterior extension of the thicker, sediment-filled gut tract and is not directly connected to the putative segmental nerve. We added labels to this part of the gut tract in Figure 1 and 2.

This questionable pygidial nerve in MCZ 1811 is roughly comparable in width and length (relative to the pygidium) to the nerves preserved under the pygidium of USNM 305903, hence our preferred, though tentative interpretation as a nerve, rather than digestive tissues. Importantly, it is not possible to determine the thickness of the VNC in MCZ 1811 due to the gut tract and glands concealing it for most of its length. Whether it was thinner or thicker than the structure preserved in USNM 305903 is unknown.

This would indicate that the later CNS is more likely the combination of gut and CNS, like the central carbon remains in MCZ 1811.

We are confident in our interpretation of the axial structures in USNM 305903 belonging to the CNS alone, for there is no indication of preserved digestive structures in this specimen. Firstly, there is no reason to infer a superimposition of structures in this individual, since it is preserved flattened laterally. This orientation would be optimal to distinguish the V(Ventral) NC from the more dorsally located digestive tract. Even imagining superimposed gut tract and VNC, one would expect these two structures to follow different courses in the anterior and posterior regions of the body – the gut tract bending towards the ventral surface at both ends, where mouth and anus were located. Instead, the whole reflective strand splits into two threads that both clearly connect to the eyes anteriorly, whereas it abruptly stops posteriorly, where the posteriormost pair of lateral threads (i.e. segmental nerves) insert. Lastly, the long and delicate (thread-like) lateral extensions do not resemble digestive glands, which are large, rounded structures broadly inserted on the gut tract in *Mollisonia* (see Fig. 1 and especially Sup. Fig. 4).

4) It is really hard to believe that the guy of mollisoniids can reach the width as indicated in Fig. 4C. It looks more like the body width under the tergites. Supporting data are needed for such a claim.

JOH: We agree with reviewer's comment. We modified Figure 4 to depict a more slender digestive system more in line with the structures observed in MCZ 1811 (Figure 1) and USNM 57663 (Supplementary Figure 4).

5) In Fig 2. Interpretative labelings are needed to help understand the authors' claims

JOH: We have included labels to clarify the morphological identity of the structures preserved and highlighted through elemental mapping.

6) As the authors notified, mollisoniids are widely accepted as in stem chelicerates. I am not sure whether it is right to exaggerate **the significance to euarthropods**, given that some detailed neuroanatomy are not that complete and informative.

JOH:

We believe that our new data convincingly demonstrate that mollisoniids illustrate a plesiomorphic condition regarding the organization of the CNS in chelicerates. As explained above, this main conclusion of our work is based on the less questionable features exhibited by the fossils, and is now further supported by our phylogenetic analysis. Because of this plesiomorphic condition within Chelicerata, we believed that our new data on mollisoniids was also relevant to the Euarthropoda as a whole, in the sense that they may help constrain reconstructions of the CNS in the last common ancestor of all crown-group euarthropods. Following comments from both referees, we now realize that discussing every implications of our new results may be confusing to the readers, and therefore counterproductive. Accordingly, we have decided to limit our discussion on the evolutionary significance of our data to the context of total-group Chelicerata, which contributes to the shortening of the discussion, to change the title of our manuscript to reflect this re-focus.

To address this comment, we have rephrased the title to make it clear that the discussion is directly focused on understanding the nervous system evolution of chelicerates, rather than euarthropods more broadly, and we have modified the abstract and discussion accordingly to emphasize these points and integrate the results of the phylogenetic analysis to better support our claims (Supplementary Figure 5).

In summary, this is an additional interesting case of fossil nerves in Cambrian arthropods. The findings will further motivate the discussion about the early evolution of arthropods, from the perspective of neuroanatomy. I would recommend to publish this paper if the authors can address the questions raised above.

We thank our colleague for his favorable recommendation.

Peiyun Cong
YKLP, Yunnan University

Reviewer #2 (Remarks to the Author):

Neural structures are rarely preserved in the fossil record, so despite the increasing reports of exceptionally preserved neuroanatomy from Cambrian panarthropod fossils in the past few years, there is still a huge amount we don't know. The manuscript by Ortega-Hernández and his co-authors described the central nervous system (CNS) of the Cambrian arthropod *Mollisonia symmetrica* from the Burgess Shale. The authors carried out careful observations, detailed descriptions, and the manuscript is well-written and well-illustrated. Therefore, this study is certainly a welcome addition to the new research field of "neuropalaeontology", and the evidence and discussion in the manuscript is generally robust and sound.

JOH: We thank the reviewer for the positive commentary on this study.

However, I have a few concerns.

1) I have my reservations regarding the optical neuropils of *Mollisonia*. The number and position of optical neuropils are very important characters for distinguishing mandibulate-like or chelicerate-like brains. In my opinion, the evidence from ROMIP 62978 in Ref [32] convincingly shows that *Mollisonia* has three optic neuropils nested outside the central brain, and the outline of two optic neuropils can also be recognised in specimen MCZ 1811 (Extended Data Fig. 5 in Ref [32]; and Fig. 1 and S1 in this manuscript), so such evidence shouldn't be dismissed easily. Therefore, the single dark area of preservation within the eye stalk in other specimens might be the result of an overlaying of retina and optic neuropils.

JOH: We have rephrased these two sentences of our discussion to clarify our position concerning the presence of neuropils in *Mollisonia*: we do not deny that Aria and Caron (2019) convincingly documented the presence of three optic neuropils in *Mollisonia plenovenatrix*, but simply state that we could not find evidence of such structures in *Mollisonia symmetrica*.

Despite further inspection of our high-resolution photographs and SEM data, we still disagree with both Aria & Caron (2019, ext. data fig. 5e) and referee 2 about their claim concerning the presence of discrete structures (neuropils) within the eye of MCZ 1811.

Reprinted by permission from Springer Nature: Nature, A middle Cambrian arthropod with chelicerae and proto-book gills, Aria and Caron, COPYRIGHT 2019

Importantly, we have opted for the most conservative coding with regard to this character in our phylogenetic analysis: the number of optic neuropils was coded as three for *M. plenovenatrix* and as uncertain (?) for *M. symmetrica*. Yet, the results of our analyses strongly support the position of both *Mollisonia* species within the chelicerate stem-lineage (see Supplementary Figure S5; Supplementary Note and Supplementary Data). In other words, this character has no major impact on the position of mollisoniids within total-group Chelicerata.

2) The arguments in the manuscript are somehow circular or even conflicting. The author cautiously emphasized several times that the poor preservation of the anterior CNS of *Mollisonia* causes uncertainty regarding the detailed features of the CNS (e.g. optic neuropil discussion above). If so, the discussion about the phylogeny based on the CNS will be built on a weak foundation, and hence there is no real evidence of conflicting phylogenetic signals between appendicular or neuroanatomical data.

JOH: Our new data on *M. symmetrica* allows a confident reconstruction of the following elements of the mollisoniid CNS: there is a single pair of lateral eyes (unlike in many chelicerates), each with a retina and a cornea; the eyes are connected to the VNC by optic nerves; the VNC extends from the posterior two-thirds of the head to the anterior two-thirds of the pygidium, and includes 11 pairs of ganglia in the trunk — a disposition that perfectly mirrors the dorsal segmentation (unlike in most chelicerates) — which are associated with thread-like segmental nerves. Only two aspects of the CNS are not well-established using our new data: the number of neuropils and the presence of a synganglion in the cephalic region.

The presence of neuropils is not confirmed in *M. symmetrica*, but it remains highly likely considering their presence in *M. plenovenatrix*. More importantly, mollisoniids are still recovered as basal stem-chelicerates, even if this uncertainty is considered in the coding of this character. As to the synganglion, it occurs in both Megacheirans and Euchelicerates and therefore, its absence in *Mollisonia* would only provide further support to our claim that mollisoniid CNS is plesiomorphic within total-group Chelicerata.

The presence of chelicerae (simple claws) in *Mollisonia*, but not in megacheirans would suggest a more crown-ward position of the latter compared to the former. Neuroanatomical data suggest otherwise, hence the mention to a conflict between neuroanatomical and appendicular data. The fact that we cannot explain this conflict does not make it less real. At this stage, we can only note that mosaic evolution is pervasive in many other metazoan groups.

For the same reason, it is not clear about the what is being illuminated with regards to the ancestral nervous system of crown-group Euarthropoda. The author can change the title and tone down some statements in the Discussion.

We have followed the reviewer's suggestion in changing the title and discussion, so that they are focused on the early evolution of the chelicerate nervous system. We have incorporated a new phylogenetic analysis that includes all instances of palaeoneuroanatomical information for Cambrian fossils (see below).

3) The phylogenetic position of *Mollisonia* has been controversial among different previous studies, which are all based on appendicular data. Neuroanatomic characters are often very useful for solving deep phylogeny. Therefore, I wonder if the authors can carry out new phylogenetic analysis with the addition of new neural characters to address this controversy? If so, some of my suggestions above would be solved too.

JOH: We have followed the reviewer suggestion and implemented a new phylogenetic analysis of 54 taxa and 106 morphological characters that incorporates paleoneuroanatomical data on Cambrian euarthropods. We have utilized an updated version of the dataset by Yang et al. (2016), which also investigated nervous system evolution in the Cambrian. We have incorporated all the relevant characters into the phylogenetic analysis (e.g. optic neuropils, extended oesophageal foramen, eye doublets, synganglion, posterior longitudinal connectives) into the analysis, and scored them as based on the available fossil data. In line with current practices in the field, we have analyzed our dataset under both maximum parsimony (equal and implied weights), as well as Bayesian inference.

We find that the results of the analyses support our interpretation that *Mollisonia* occupies an earlier branching position within the chelicerate stem-lineage relative to megacheirans, based on the neuroanatomical data available (see Supplementary Figure 5). We believe that this analytical support strengthens our conclusions, and provides a more compelling case for our discussion on the conflict between external morphology and fossilized neuroanatomy.

A few minor points below:

1) There is some repetition in the manuscript, such as preservation in the Results and in the Discussion, which could be consolidated.

JOH: We have removed instances of redundant information to make the Discussion more streamlined, including relocating the section on preservation as part of the Supplementary Information.

2) Line 405: (Figure S5)--There is no figure.

JOH: Thanks for spotting this typo. Corrected to Supplementary Figure 1.

3)In Fig. 3 caption: “ssn” should be “sn”

JOH: Corrected to *snn*, for segmental nerve number.

4)Fig. S2 is identical to Fig. 3, apart from the figure caption

JOH: That is correct. Our intention is to provide the reader with a clean view of the fossil material itself, so that it is possible to observe the data as clearly as possible.

Cited references:

- Aria, C. and Caron, J.B., 2019. A middle Cambrian arthropod with chelicerae and proto-book gills. *Nature*, 573(7775), pp.586-589.
- Butterfield, N.J., 2002. *Leaenchoilia* guts and the interpretation of three-dimensional structures in Burgess Shale-type fossils. *Paleobiology*, 28(1), pp.155-171.
- Chen, J., Waloszek, D. and Maas, A., 2004. A new ‘great-appendage’ arthropod from the Lower Cambrian of China and homology of chelicerate chelicerae and raptorial antero-ventral appendages. *Lethaia*, 37(1), pp.3-20.
- Haug, J.T., Waloszek, D., Maas, A., Liu, Y.U. and Haug, C., 2012. Functional morphology, ontogeny and evolution of mantis shrimp-like predators in the Cambrian. *Palaeontology*, 55(2), pp.369-399.
- Legg, D.A., Sutton, M.D. and Edgecombe, G.D., 2013. Arthropod fossil data increase congruence of morphological and molecular phylogenies. *Nature communications*, 4(1), pp.1-7.
- Lerosey-Aubril, R., Hegna, T.A., Kier, C., Bonino, E., Habersetzer, J. and Carré, M., 2012. Controls on gut phosphatisation: the trilobites from the Weeks Formation Lagerstätte (Cambrian; Utah). *PLoS One*, 7(3), p.e32934.
- Liu, Y., Hou, X.G. and Bergström, J., 2007. Chengjiang arthropod *Leaenchoilia illecebrosa* (Hou, 1987) reconsidered. *Gff*, 129(3), pp.263-272.
- Liu, Y., Ortega-Hernández, J., Zhai, D. and Hou, X., 2020. A reduced labrum in a Cambrian great-appendage euarthropod. *Current Biology*, 30(15), pp.3057-3061.
- Tanaka, G., Hou, X., Ma, X., Edgecombe, G.D. and Strausfeld, N.J., 2013. Chelicerate neural ground pattern in a Cambrian great appendage arthropod. *Nature*, 502(7471), pp.364-367.
- Vannier, J., Liu, J., Lerosey-Aubril, R., Vinther, J. and Daley, A.C., 2014. Sophisticated digestive systems in early arthropods. *Nature communications*, 5(1), pp.1-9.

REVIEWERS' COMMENTS

Reviewer #1 (Remarks to the Author):

I am happy with the authors response and the revised version of manuscript. I would recommend to publish this after a checking though of spelling-erros, such as in line 231-231, 'in more recent times' should be 'in more recent time'.

Reviewer #2 (Remarks to the Author):

I appreciate that the authors have taken many of my previous suggestions into consideration, including carrying out additional phylogenetic analyses and edited the discussion and title, so the manuscript is now significantly improved from its previous version. I am happy with most description and discussion in this manuscript, and a lot of speculative discussion has been removed. The phylogenetic position of Mollisonia as one of the most basal branches within the stem-group chelicerates seem to be sound, and the interpretation on the neuroanatomy innovation on the early chelicerates is very interesting and significant. Therefore, I recommend this manuscript to be published in Nature Communications.

REVIEWERS' COMMENTS

Reviewer #1 (Remarks to the Author):

I am happy with the authors response and the revised version of manuscript. I would recommend to publish this after a checking though of spelling-erros, such as in line 231-231, 'in more recent times' should be 'in more recent time'.

JOH: We thank the reviewer for the positive and constructive comments that helped us improve this study, and have screened the manuscript to identify spelling errors.

Reviewer #2 (Remarks to the Author):

I appreciate that the authors have taken many of my previous suggestions into consideration, including carrying out additional phylogenetic analyses and edited the discussion and title, so the manuscript is now significantly improved from its previous version. I am happy with most description and discussion in this manuscript, and a lot of speculative discussion has been removed. The phylogenetic position of Mollisonia as one of the most basal branches within the stem-group chelicerates seem to be sound, and the interpretation on the neuroanatomy innovation on the early chelicerates is very interesting and significant. Therefore, I recommend this manuscript to be published in Nature Communications.

JOH: We thank the reviewer for their comments that helped us improve the manucript for publication.